# High Speed Decoding for High-Rate and Short-Length Reed–Muller Code Using Auto-Decoder

Hyun Woo Cho and Young Joon Song *

Department of Electronic Engineering, Kumoh National Institute of Technology, Gumi 39177, Korea
* Correspondence: yjsong@kumoh.ac.kr

**Abstract:** In this paper, we show that applying a machine learning technique called auto-decoder (AD) to high-rate and short length Reed–Muller (RM) decoding enables it to achieve maximum likelihood decoding (MLD) performance and faster decoding speed than when fast Hadamard transform (FHT) is applied in additive white Gaussian noise (AWGN) channels. The decoding speed is approximately 1.8 times and 125 times faster than the FHT decoding for $\Re(1,4)$ and $\Re(2,4)$, respectively. The number of nodes in the hidden layer of AD is larger than that of the input layer, unlike the conventional auto-encoder (AE). Two ADs are combined in parallel and merged together, and then cascaded to one fully connected layer to improve the bit error rate (BER) performance of the code.

**Keywords:** Reed–Muller (RM) code; machine learning; auto-decoder; auto-encoder

## 1. Introduction

Machine learning (ML) techniques are widely used in many fields, such as image recognition, natural language processing, and autonomous driving [1–3]. Auto-encoder (AE) unsupervised ML techniques are known for their capacity to extract important features of data while reducing unwanted noise, and thus are useful in generating new images with key features [4]. AEs perform roles such as dimensionality reduction, image denosing, image generation, and abnormality detection, and are used in various fields such as medical care, autonomous driving, and image recognition [5–8]. In addition, various studies are being undertaken to apply machine learning technology to communication systems, such as channel coding, massive multi-input and multi-output, multiple access, resource allocation, and network security [9,10]. In this study, we modify an AE to a new model called the auto-decoder (AD), which is suitable for reducing the noise that corrupts the transmitted information signal in channel coding. The proposed AD is used to decode Reed–Muller (RM) code of high-rate and short length, which is used in many communication systems, such as long-term evolution (LTE) and fifth-generation wireless (5G) cellular systems [11,12], where the minimum latency delay is 5 ms. The requirement is to further reduce this delay in sixth-generation (6G) wireless systems [13,14]. Since we consider high-rate Reed–Muller code of short length, such as $\Re(2,4)$ with code rate 0.6875, we use a fast Hadamard transform (FHT) decoding method [15] for performance comparison instead of the recursive decoding of [16,17] which is useful for low-rate RM code. Because the RM code has an extremely simple structure and can be decoded with maximum likelihood decoding (MLD) performance using FHT, it is especially useful in control channels in wireless communication systems. We first illustrate the key differences between the conventional AE and the proposed AD, and then show how to construct the decoder for the RM code using it. After training the AD model for the code, we found that the proposed method showed similar performance to the conventional FHT method with faster decoding speed. For improved performance, we present a parallel auto-decoder (PAD) that combines a couple of ADs in parallel.

## 2. RM Decoder Based on AD

This section explains the RM decoder using AD and compares the performance of FHT decoding. Table 1 shows the notations used in this paper.

**Table 1.** summarizes the notation used in the paper.

| Symbol | Description |
|---|---|
| $n$ | length of codeword (bits) |
| $r, m$ | parameters of RM code ($0 \leq r \leq m$) |
| $k$ | length of message (bits) |
| $N$ | number of nodes in FC layer |
| **y** | one-hot encoding vector |
| **m** | message vector |
| **z** | output of FC layer |
| $S$ | number of validation sets |
| $\rho_t, \rho_{v,s}$ | SNRs for the training set and the $s$-th validation set |

### 2.1. Auto-Decoder

The AD, which plays a central role in decoding of the RM code, is modified from a conventional AE. Figure 1 shows the basic structures of the conventional AE and the proposed AD, highlighting the difference between the number of nodes in two different hidden layers. The number of nodes in the hidden layer of the AE in Figure 1a is less than that of the input layer. In comparison to the AE, the number of nodes in the hidden layer of the AD in Figure 1b is larger than that of the input layer. Figure 2 shows the typical structure of an AD that is composed of three hidden layers for the decoder; the number of nodes in each layer is presented in Table 2. The number of nodes in the input layer is the same as the length of the codeword $n$. The number of nodes in the first hidden layer is $2n$, in the second hidden layer is $4n$, in the last hidden layer is again $2n$, and finally the output layer becomes $n$ again. In general, a neural network with a multilayer perceptron shows much better performance than a single-layer perceptron. From this perspective, we can speculate that the hidden layer structure of the AD is more suitable for decoding short length codes, such as RM code, than the AE, in which the number of nodes of the hidden layer becomes smaller as the depth of the hidden layer increases, which may result in a smaller number of nodes, especially in the middle of the hidden layer, resulting in poor decoding performance.

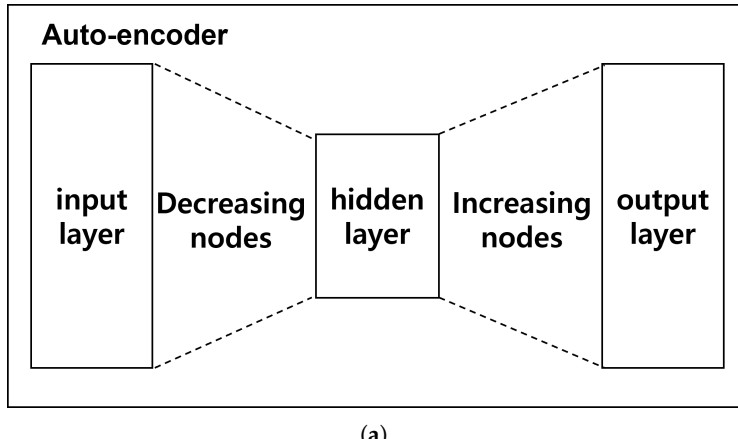

(a)

**Figure 1.** *Cont.*

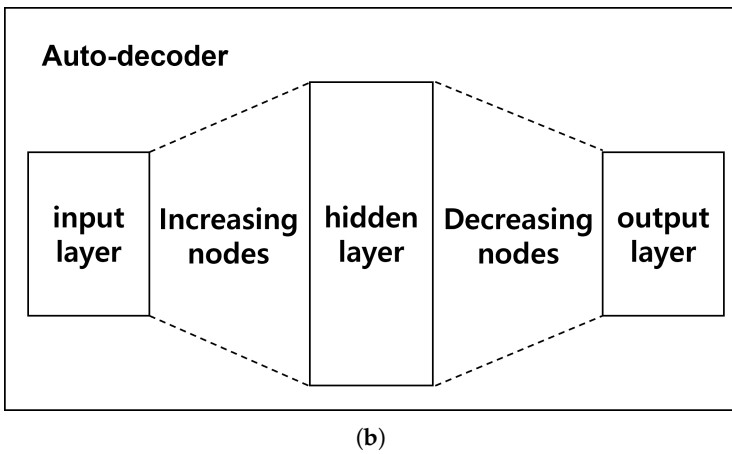

(**b**)

**Figure 1.** Structure of (**a**) auto-encoder and (**b**) auto-decoder.

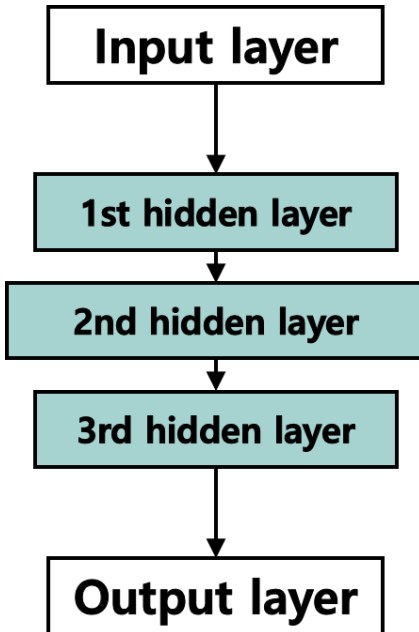

**Figure 2.** Structure of auto-decoder with 3 hidden layers.

**Table 2.** Number of nodes for layers in AD.

| Layer | Number of Nodes |
|---|---|
| input | $n$ |
| 1st hidden | $2n$ |
| 2nd hidden | $4n$ |
| 3rd hidden | $2n$ |
| output | $n$ |

### 2.2. RM Decoding Model

The RM code of $n = 2^m$ bits with the minimum Hamming distance of $2^{m-r}$ used to encode $k = \sum_i^r \binom{m}{i}$ bits of message is denoted as $\mathfrak{R}(r, m)$ [15,18]. To illustrate the proposed decoding model, we consider two cases of RM code, $\mathfrak{R}(1, 4)$ and $\mathfrak{R}(2, 4)$ of code length 16, because the model training for longer code consumes more time. Figure 3 shows the proposed RM decoder structure constructed in such a way that the AD of Figure 2 is followed by one fully connected (FC) layer with the number of nodes defined as $N = 2^k$, and, thus, $N = 2^5$ for $\mathfrak{R}(1, 4)$ and $N = 2^{11}$ for $\mathfrak{R}(2, 4)$. Because $N = 2^k$ is equivalent to the number of all possible messages transmitted, we use the FC layer as the output layer.

Moreover, a softmax activation function is used to normalize the output of the model to a probability distribution over all possible transmitted messages. Figure 4 shows the method used to estimate the transmitted message bits of the (16,11) code of $\mathfrak{R}(2,4)$ from the output layer. The output node index of the FC layer is the decimal value corresponding to the message bits, and the transmitted message bits are estimated by converting the index of the maximum output node value into $k = 11$ binary bits.

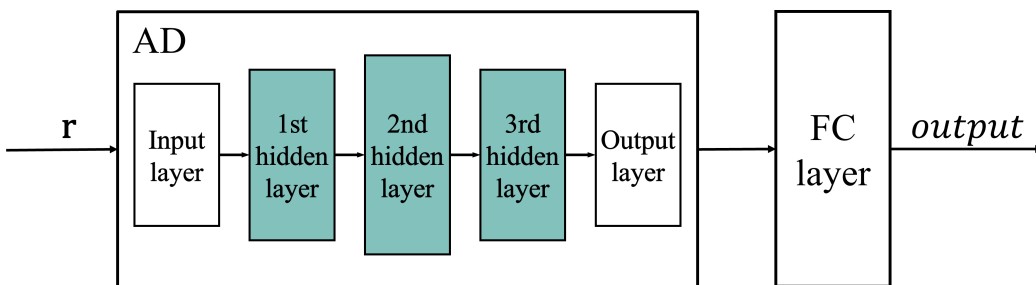

**Figure 3.** Structure of RM decoder based on AD.

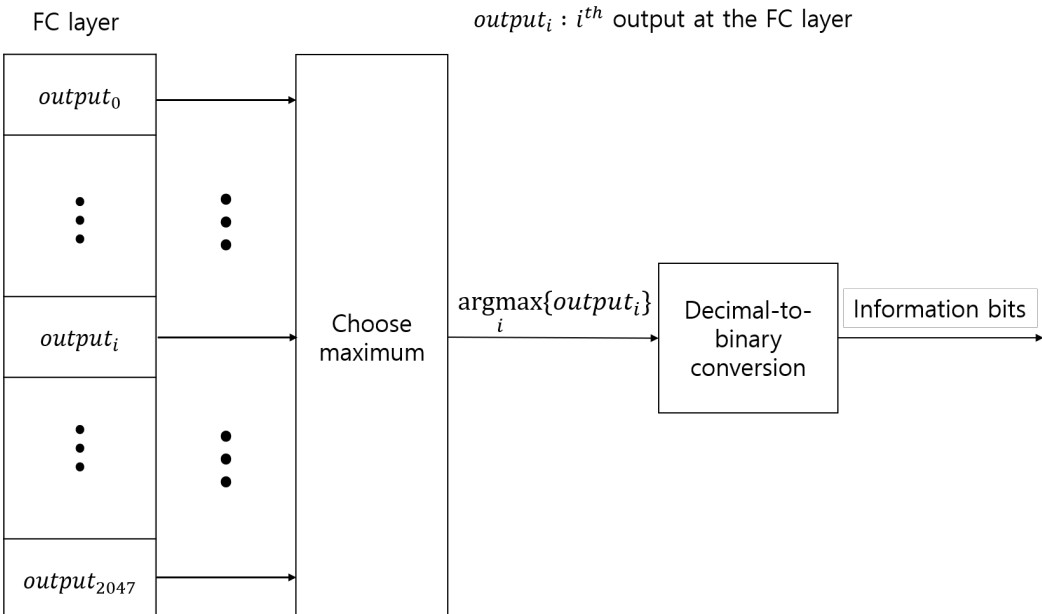

**Figure 4.** Message estimation of $\mathfrak{R}(2,4)$ from the output of FC layer.

*2.3. Hyperparameters*

Table 3 lists the hyperparameters used for training the RM decoder. Let $\mathbf{y} = \{0,1\}^N = (y_0, y_1, \cdots, y_j, \cdots, y_{N-1}) = (0, 0, \cdots, 1, \cdots, 0)$ be the one-hot encoded vector for the message $\mathbf{m} = (m_0, m_1, \cdots, m_k - 1)$, where $j = \sum_{l=0}^{k-1} m_l 2^l$ is the decimal value for $\mathbf{m}$, and all bits of $\mathbf{y}$ are zeros, except the value of one at the $j$-th bit. Let $z_i$ be the $i$-th output of the FC layer. The cross-entropy is given by:

$$L(\mathbf{y}, \mathbf{z}) = -\sum_{i=0}^{N-1} [y_i \log z_i + (1 - y_i) \log(1 - z_i)] \tag{1}$$

is used as the loss function, and the Adam optimizer is used for model training. The training data set was $2^k \times 10^5$, the epoch was $10^2$, and the batch size was set to $10^4$. Normalized validation error (NVE) of [19]

$$\text{NVE} = \frac{1}{S} \sum_{s=1}^{S} \frac{BER_{AD}(\rho_t, \rho_{v,s})}{BER_{FHT}(\rho_{v,s})} \tag{2}$$



is used to select the appropriate signal-to-noise (SNR) ratio for model training, where $\rho_t$ and $\rho_{v,s}$ are the SNRs for the training set and the $s$-th validation set, respectively, and $S$ is the number of all possible different validation sets. $BER_{AD}(\rho_t, \rho_{v,s})$ is the bit error rate when the RM decoder with AD is trained at $\rho_t$ and evaluated at $\rho_{v,s}$. Table 4 shows the NVE values at different training SNRs, from 0 dB to 7 dB with 1 dB intervals, and we observe that NVE = 0.945 at $\rho_t = 1$ dB is the lowest value among them. Thus, the training SNR is set to 1 dB.

**Table 3.** Hyperparameters for model training.

| | |
|---|---|
| loss function | cross-entropy |
| optimizer | Adam |
| training data set | $2^k \times 10^5$ |
| epoch | $10^2$ |
| batch size | $10^4$ |

**Table 4.** NVE for different training SNRs.

| Training SNR ($\rho_t$) | 0 | 1 | 2 | 3 |
|---|---|---|---|---|
| NVE | 0.972 | 0.945 | 0.982 | 1.138 |
| Training SNR ($\rho_t$) | 4 | 5 | 6 | 7 |
| NVE | 0.981 | 1.320 | 1.529 | 2.489 |

*2.4. Performance Evaluation*

To evaluate the performance of the proposed decoder using AD, we consider two cases of RM code, $\mathfrak{R}(1,4)$ and $\mathfrak{R}(2,4)$ whose code structure is simple, with a short message length. Figure 5 shows the BER of RM coding with the AD compared with FHT decoding. The BER at each SNR was calculated when the maximum number of error bits was 500, or the number of codewords generated reached $10^5$. From the graph, we can see that the two methods show almost the same BER performance. Table 5 shows the decoding times for $\mathfrak{R}(1,4)$ and $\mathfrak{R}(2,4)$ using FHT decoding and the AD. A computer with a central processing unit (CPU) of Intel i9-7920, graphics processing unit (GPU) of Nvidia Titan XP, and 64 GB random access memory (RAM) was used for the evaluation. If a GPU is used for decoding, the proposed method can have an advantage over the method using FHT, and for a fair comparison, we measure the decoding time using a CPU without a GPU. The decoding time of the proposed method was 1.8 times faster than the decoding time of FHT decoding for $\mathfrak{R}(1,4)$ and 125 times faster for $\mathfrak{R}(2,4)$. This derives from the fact that (16,11) $\mathfrak{R}(2,4)$ decoding using FHT needs $2^6$ FHTs, considering six masking bits, whereas (16,5) $\mathfrak{R}(1,4)$ decoding needs only one FHT operation [15]. The main difference between the RM decoding using the AD for $\mathfrak{R}(1,4)$ and $\mathfrak{R}(2,4)$ is the number of nodes in the FC layer, which increases the number of parameters.

**Table 5.** Decoding time of RM code using FHT and AD.

| RM Code | Method | Time (ms) |
|---|---|---|
| $\mathfrak{R}(1,4)$ | FHT | 0.6012 |
| | AD | 0.3327 |
| $\mathfrak{R}(2,4)$ | FHT | 46.625 |
| | AD | 0.3704 |

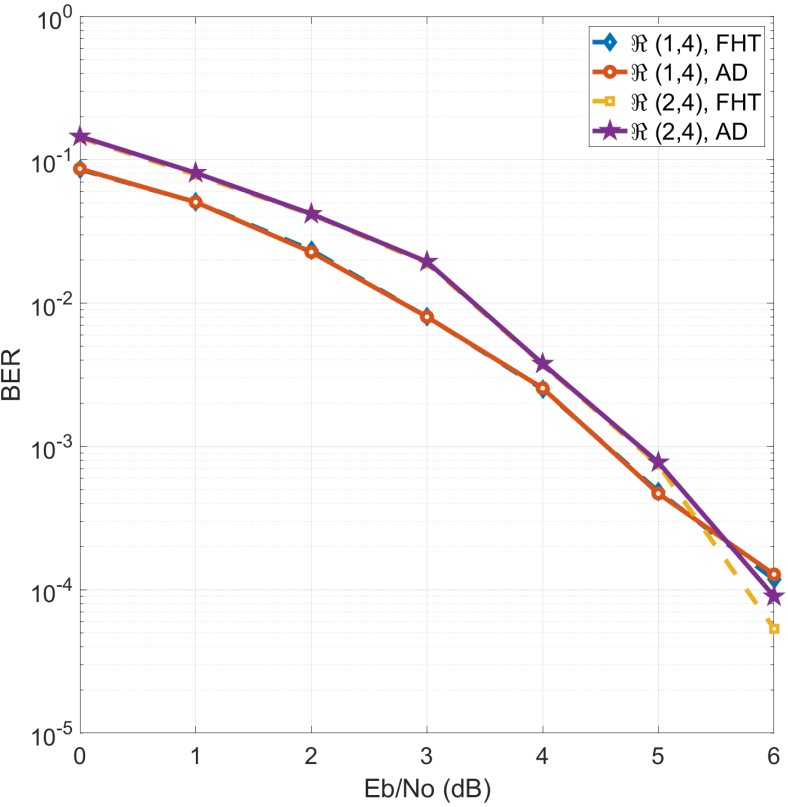

**Figure 5.** BER of RM decoding with FHT and AD.

## 3. PAD

To improve the BER performance of the RM decoder using a single AD, we present a PAD composed of multiple ADs. We call an AD in a PAD a constituent auto-decoder (CAD). To illustrate the structure of a PAD, we set the number of CADs in the PAD to five, as shown in Figure 6, where the output layers of all CADs are simply added at the merge layer with $n = 2^m$ nodes. Each CAD has the same structure, except for the different number of nodes at the first, second, and third hidden layers, as described in Table 6, where the first and the third hidden layers have the same number of nodes; thus, we find the symmetry structure of the PAD based on the second hidden layer or middle layer. As in the case of the AD, the activation functions for the hidden and output layers of CADs in the PAD are exponential linear unit (ELU) and tanh functions, respectively [20,21]. The PAD is followed by the FC layer, and the hyperparameters for the decoder model using PAD are the same as in the case of the AD. Figure 7 shows the BER performance for $\mathfrak{R}(1, 4)$ and $\mathfrak{R}(2, 4)$ using the conventional FHT decoder, and the proposed decoder model denoted as PAD-*i*, where *i* CADs are used. The specifications of the computer and the conditions for the BER calculation in Figure 7 for PAD are the same as those used in Figure 5 for the AD. Figure 7a shows the BER for $\mathfrak{R}(1, 4)$ in the range from 0 to 6 dB with 1 dB steps, and (b) in the range of 1.5 dB to 2.5 dB, which provides a better discrimination between different cases where no significant difference can be observed in terms of BER. However, PAD-2 shows the best performance. Figure 7c shows the BER for $\mathfrak{R}(2, 4)$ in the range from 0 to 6 dB with a 1 dB step, and (d) in the range of 1.5 dB to 2.5 dB for better observation. PAD-3 demonstrates the best performance, but there is no significant improvement as we increase the number of CADs in PAD. The comparison of decoding process times and the number of parameters for $\mathfrak{R}(1, 4)$ and $\mathfrak{R}(2, 4)$ using FHT and PADs is described in Table 7, where the higher the number of CADs in PAD, the higher the number of parameters and the complexity, and the longer the decoding process time. In $\mathfrak{R}(1, 4)$, the number of parameters of PAD-1 and PAD-5 are about 10-fold different, but the decoding time is only about 1.3-fold different. This is because the computation of CADs is performed in parallel.

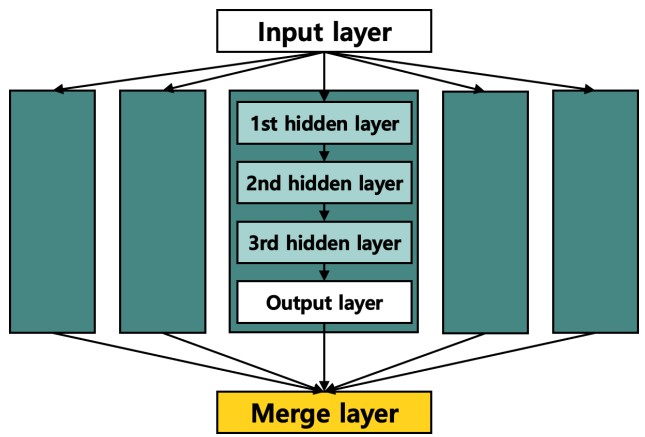

**Figure 6.** Structure of PAD with 5 CADs.

**Figure 7.** BER of RM decoding with FHT and PADs.

**Table 6.** The number of nodes in PAD with 5 CADs.

| Layer | Number of Nodes | | | | |
|---|---|---|---|---|---|
| | 1st CAD | 2nd CAD | 3rd CAD | 4th CAD | 5th CAD |
| 1st hidden | $n$ | $2n$ | $3n$ | $4n$ | $5n$ |
| 2nd hidden | $n$ | $4n$ | $9n$ | $16n$ | $25n$ |
| 3rd hidden | $n$ | $2n$ | $3n$ | $4n$ | $5n$ |
| output | $n$ | $n$ | $n$ | $n$ | $n$ |

**Table 7.** RM decoding time using FHT and PADs.

| Method | Time (ms) | | Parameters | |
|---|---|---|---|---|
| | $\mathfrak{R}(1,4)$ | $\mathfrak{R}(2,4)$ | $\mathfrak{R}(1,4)$ | $\mathfrak{R}(2,4)$ |
| FHT | 0.6012 | 46.625 | - | - |
| PAD-1 | 0.3327 | 0.3704 | 5808 | 40,080 |
| PAD-2 | 0.3472 | 0.3785 | 6896 | 41,168 |
| PAD-3 | 0.3838 | 0.4037 | 22,512 | 56,784 |
| PAD-4 | 0.4075 | 0.4367 | 57,728 | 92,000 |
| PAD-5 | 0.4520 | 0.4854 | 124,864 | 159,136 |
| PAD-6 | 0.4972 | 0.5241 | 239,312 | 273,584 |
| PAD-7 | 0.5508 | 0.6225 | 419,536 | 388,032 |
| PAD-8 | 0.6198 | 0.6352 | 687,072 | 502,480 |

## 4. Conclusions

In this paper, we proposed $\mathfrak{R}(1,4)$ and $\mathfrak{R}(2,4)$ decoders using an AD, with MLD performance and shorter decoding process time than the FHT decoder. The decoding time of the RM decoder using the AD is 1.8 times faster than that using FHT decoding for $\mathfrak{R}(1,4)$, and 125 times faster for $\mathfrak{R}(2,4)$. This is because the FHT decoding method relies heavily on masking bits. We presented PAD with multiple CADs to improve the BER performance of the RM decoder using a single AD, and found that PAD-2 and PAD-3 showed the best performance for $\mathfrak{R}(1,4)$ and $\mathfrak{R}(2,4)$, respectively; however, the performance difference is not significant. The proposed fast decoding method with MLD performance can be useful in mobile communication systems, such as 5G and 6G, which require low latency and BER. Since the AD shows better performance then the AE when input size is small, it can be useful for noise removal in signal processing with a relatively small data size. The AD can be used not only in communication fields, but also in fields using various sensors. As we have confirmed the performance of the proposed model based on the AD and PAD for high-rate and short-length RM codes in terms of decoding speed and BER, we will extend the result to other error-correction coding schemes.

**Author Contributions:** Software, H.W.C.; Writing—review & editing, Y.J.S. All authors have read and agreed to the published version of the manuscript.

**Funding:** This work was supported by the National Research Foundation of Korea (NRF) grant funded by the Korea government (MSIT) (No. 2021R1F1A1061907).

**Institutional Review Board Statement:** Not applicable.

**Informed Consent Statement:** Not applicable.

**Data Availability Statement:** Not applicable.

**Conflicts of Interest:** The authors declare no conflict of interest.

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
