# Peer review of "High Speed Decoding for High-Rate and Short-Length Reed–Muller Code Using Auto-Decoder"

_applsci, doi:10.3390/app12189225_

Round 1

Reviewer 1 Report

The work addresses an interesting and current topic. I suggest the authors to introduce some real world scenarios where their result can be used according to the key terms used: Artificial Intelligence; virtualization; view information; Human interaction with the computer. Now the text only refers to a narrow audience.

What is the practical and theoretical model from real life in agreement with virtualization; view information; Human interaction with the computer that your result solves?

I mention that the results obtained in the machine learning technique called auto-decoder (AD) are necessary in all areas of life, so the paper must have a discussion section in which to argue the implications of the results obtained in some areas of life (medicine, marketing, etc.)

Author Response

Response to Reviewer 1 Comments

Point 1: The work addresses an interesting and current topic. I suggest the authors to introduce some real world scenarios where their result can be used according to the key terms used: Artificial Intelligence; virtualization; view information; Human interaction with the computer. Now the text only refers to a narrow audience.

What is the practical and theoretical model from real life in agreement with virtualization; view information; Human interaction with the computer that your result solves?

Response 1: As suggested by the Reviewer, we have modified the introduction as follows:

In Introduction (from line 15-22 of page 1):

AE performs roles such as dimensionality reduction, image denosing, image generation, and abnormality detection, and is being used in various fields such as medical care, autonomous driving, and image recognition [5-8]. In addition, various studies are being conducted to apply machine learning technology to communication systems such as channel coding, massive multi-input and multi-output, multiple access, resource allocation, and network security [9,10]. In this study, we modify AE to a new model called auto-decoder (AD), which is suitable for reducing the noise that corrupts the transmitted information signal in channel coding.

Point 2: I mention that the results obtained in the machine learning technique called auto-decoder (AD) are necessary in all areas of life, so the paper must have a discussion section in which to argue the implications of the results obtained in some areas of life (medicine, marketing, etc.)

Response 2: As suggested by the Reviewer, we have modified the conclusion as follows:

In Conclusion (from line 145-147 of page 7-8):

Since AD shows better performance then AE when input size is small, it can be useful for noise removal in signal processing with a relatively small data size. AD can be used not only in communication fields but also in fields using various sensors.

Reviewer 2 Report

The reviewer found the contribution of the work is not significant. In the existing literature, researchers used a similar concept/design for non-orthogonal multiple access systems. 

1. Figure 7 is hard to read the curves;

2. In the First sentence of the Conclusion Section, the author does not propose RM code using AD. 

Author Response

Response to Reviewer 2 Comments

Point 1: The reviewer found the contribution of the work is not significant. In the existing literature, researchers used a similar concept/design for non-orthogonal multiple access systems. 

Response 1: Non-orthogonal multiple access systems is a multiple access method that allows non-orthogonal resources to be allocated by sharing frequency, time, and code resources for each user terminal, while accommodating a larger number of terminals with a set resource. However, our proposed method is the RM decoding method related to channel coding and is not related to non-orthogonal multiple access systems.

Point 2: Figure 7 is hard to read the curves.

Response 2: We agree with the Reviewer’s comment. However, we have already added a graph with an enlarged curve to figure 7 (In Section 3 (form line 122 – 127 of page 7):

Fig.7 (a) shows the BER for R(1,4) in the range from 0 to 6 dB with 1 dB steps, and (b) in the range of 1.5 dB to 2.5 dB, which provides a better observation between different cases where no significant difference can be observed in terms of BER. However, PAD-2 shows the best performance. Fig.7 (c) shows the BER for R(2,4) in the range from 0 to 6 dB with a 1 dB step, and (d) in the range of 1.5 dB to 2.5 dB for better observation.).

Point 3: In the First sentence of the Conclusion Section, the author does not propose RM code using AD. 

Response 3: We agree with the Reviewer’s comment. We have modified conclusion as follows:

In Conclusion (from line 136 of page 7): 

In this paper, we proposed R(1,4) and R(2,4) decoders using AD, with MLD performance and shorter decoding process time than the FHT decoder.

Reviewer 3 Report

The authors focus their study on applying a machine learning technique called auto-decoder to high-rate and short length Reed-Muller decoding in order to enable it to achieve maximum likelihood decoding  performance and faster decoding speed than when fast Hadamard transform is applied in additive white Gaussian noise channels. The manuscript is overall well written and easy to follow and the authors have well thought out their main contributions. The provided theoretical analysis is concrete, complete, and correct and the authors have provided all the intermediate steps in order to enable the average reader to easily follow it. Furthermore, the provided numerical results are also rich in order to show the pure operation and the performance of the proposed framework. The authors are encouraged to address the following comments provided by the reviewer in order to improve the scientific depth of their manuscript, as well as they need to consider the following suggestions in order to improve the quality of presentation of the manuscript. Initially, the provided related work in section one is extremely limited for a journal paper publication and the authors need to better summarize the state-of-the-art. The authors need to provide some background information about machine learning techniques, such as Huang, Xin-Lin, Xiaomin Ma, and Fei Hu. "Machine learning and intelligent communications." Mobile Networks and Applications 23.1 (2018): 68-70, that have been introduced in the literature in similar application scenarios. This will help the average leader to easily follow the rest of the analysis. In Section 2, the authors need to include a table summarizing the main notation that has been used in the paper and provide the corresponding units wherever this is appropriate. In Section 3, the authors need to include an additional subsection providing the theoretical analysis of the computational complexity of the proposed approach. Based on the previous comments some indicative numerical results capturing the computational complexity of the proposed method need to be provided. Finally, the overall  manuscript needs to be checked for typos, syntax, and grammar errors in order to improve the quality of its presentation.

Author Response

Response to Reviewer 3 Comments

Point 1: Initially, the provided related work in section one is extremely limited for a journal paper publication and the authors need to better summarize the state-of-the-art. The authors need to provide some background information about machine learning techniques, such as Huang, Xin-Lin, Xiaomin Ma, and Fei Hu. "Machine learning and intelligent communications." Mobile Networks and Applications 23.1 (2018): 68-70, that have been introduced in the literature in similar application scenarios. This will help the average leader to easily follow the rest of the analysis.

Response 1: We agree with the Reviewer’s comment. As suggested, we have modified the introduction as follows:

In Introduction (from line 15-20 of page 1):

AE performs roles such as dimensionality reduction, image denosing, image generation, and abnormality detection, and is being used in various fields such as medical care, autonomous driving, and image recognition [5-8]. In addition, various studies are being conducted to apply machine learning technology to communication systems such as channel coding, Massive multi-input and multi-output, multiple access, resource allocation, and network security [9,10].

Point 2: In Section 2, the authors need to include a table summarizing the main notation that has been used in the paper and provide the corresponding units wherever this is appropriate.

Response 2: As suggested by the Reviewer, we have modified the Section 2 as follows: In Section 2 (from Table 1 of page 2), we added a table summarizing the notation in the paper.

Point 3: In Section 3, the authors need to include an additional subsection providing the theoretical analysis of the computational complexity of the proposed approach. Based on the previous comments some indicative numerical results capturing the computational complexity of the proposed method need to be provided.

Response 3:

As suggested by the Reviewer, we added the number of parameters of the models in Table 7. From a computer science perspective, computational complexity analysis is almost always attributed to the Big-O notation of the algorithm. However, from the engineering standpoint, the Big-O is often an oversimplified measure that cannot be immediately translated into the hardware resources required to realize the algorithm (NNs) in a hardware platform. We used the number of parameters of the model represents the complexity of the model.

As suggested, we have modified the Section 3 as follows:

In Section 3. PAD (from line 129– 134 of page 7 and Table 7):

The comparison of decoding process times and the number of parameters for R(1,4) and R(2,4) using FHT and PADs is described in Table 7, where higher the number of CADs in PAD, the higher the number of parameters and the complexity, and the longer the decoding process time. In R(1,4), the number of parameters of PAD-1 and PAD-5 are about 10 times different, but the decoding time is only about 1.3 times different. This is because the computation of CADs is performed in parallel.

Pont 4: Finally, the overall manuscript needs to be checked for typos, syntax, and grammar errors in order to improve the quality of its presentation.

Response 4: We corrected typos, syntactic and grammatical errors in the overall manuscript.

Round 2

Reviewer 2 Report

Thanks for the improvement on the paper, especially on the figures. Yes the proposed method is not about NOMA. However, similar idea has been studied in NOMA setting with different high rate short code. See: R. Abbas, T. Huang, B. Shahab, M. Shirvanimoghaddam, Y. Li, and B. Vucetic, 2020. Grant-free non-orthogonal multiple access: A key enabler for 6G-IoT. arXiv preprint arXiv:2003.10257. The reviewer suggest to cite this related work in the background so that the literature review is more complete. 

Author Response

Point 1: Thanks for the improvement on the paper, especially on the figures. Yes the proposed method is not about NOMA. However, similar idea has been studied in NOMA setting with different high rate short code. See: R. Abbas, T. Huang, B. Shahab, M. Shirvanimoghaddam, Y. Li, and B. Vucetic, 2020. Grant-free non-orthogonal multiple access: A key enabler for 6G-IoT. arXiv preprint arXiv:2003.10257. The reviewer suggest to cite this related work in the background so that the literature review is more complete.

Response 1: As suggested by the Reviewer, we have modified the introduction as follows:

In introduction (from line 22-26 of page 1)

The proposed AD is used to decode Reed-Muller (RM) code of high-rate and short length, which is used in many communication systems, such as long-term evolution (LTE) and fifth-generation wireless (5G) cellular systems [11, 12], where the minimum latency delay is 5 msec, and the requirement is to further reduce this delay in sixth-generation (6G) wireless systems [13, 14].

Reviewer 3 Report

The authors have addressed the reviewers comments. The quality of presentation and the scientific depth of the manuscript have been substantially improved.

Author Response

Point 1: The authors have addressed the reviewers comments. The quality of presentation and the scientific depth of the manuscript have been substantially improved.

Response 1: -We would like to thank the reviewer for their useful comments and valuable suggestions, which we believe have improved the quality of the paper.
